# IL-1R8 as Pathoimmunological Marker for Severity of Canine Chronic Enteropathy

**DOI:** 10.3390/vetsci9060295

**Published:** 2022-06-14

**Authors:** Federica Riva, Laura Bianchessi, Camilla Recordati, Alessia Inglesi, Vittoria Castiglioni, Lauretta Turin

**Affiliations:** 1Department of Veterinary Medicine and Animal Sciences, University of Milan, Via dell’Università 6, 26900 Lodi, Italy; federica.riva@unimi.it (F.R.); laura.bianchessi@unimi.it (L.B.); camilla.recordati@unimi.it (C.R.); alessia.inglesi@unimi.it (A.I.); 2Idexx Laboratories Italia, Via Guglielmo Silva 36, 20149 Milan, Italy; vittoria-castiglioni@idexx.com

**Keywords:** IL-1R8, TIR8, SIGIRR, chronic enteropathy, *Helicobacter*, dog, biomarker

## Abstract

Chronic enteropathy (CE) is a severe multifactorial gastrointestinal disease that affects dogs and is driven by poorly characterized inflammatory pathways. Imbalance of pro-inflammatory response regulators, including IL-1R8, may be due to different factors, among which the infection with *Helicobacteraceae* is known to lead to a vicious circle in which excessive pro-inflammatory signaling and gastrointestinal injury reinforce each other and boost the disease. We investigated the expression of IL-1R8 in large intestine biopsies of dogs with or without clinical signs of CE and with previously assessed enterohepatic *Helicobacter* spp. colonization status by mean of quantitative real-time PCR. Our study revealed that IL-1R8 is downregulated in both acutely (*p* = 0.0074) and chronically (*p* = 0.0159) CE affected dogs compared to healthy controls. The data also showed that IL-1R8 expression tends to decrease with colonization by *Helicobacter* spp. Interestingly, a negative correlation was detected between the level of expression of IL-1R8 and the severity of macroscopic lesions identified by endoscopy and the crypt hyperplasia score. We further compared the expression levels between males and females and found no statistically significant difference between the two groups. No significant difference was observed in IL-1R8 expression profiles with the age of the animals either. Interestingly, an association was uncovered between IL-1R8 expression level and dog breed. Together, our data advance knowledge on gastrointestinal pathoimmunology in dogs and highlight the potential utilization of IL-1R8 as a diagnostic, prognostic and therapeutic biomarker for canine chronic enteropathy.

## 1. Introduction

Canine chronic enteropathy (CE) is an intestinal idiopathic inflammatory disorder which has gained considerable attention in recent years due to severity of symptoms, failure of diet-, antibiotic- and other drug-based treatments and irreversible damage outcome [1,2,3]. CE includes food-responsive, antibiotic-responsive, immunosuppressant-responsive and non-responsive cases [2]. Diagnosis of CE includes endoscopic and histologic examinations of biopsies after other all other possible causes of disease have been excluded. Therefore, the recognition of reliable biomarkers suitable for objective assessment of CE diagnosis, severity, and prognosis is advisable.

The decoy IL-1 (interleukin-1) family receptor IL-1R8 (interleukin-1 receptor 8), formerly TIR8 (Toll/interleukin-1 receptor 8) or SIGIRR (single-immunoglobulin-interleukin-1 related receptor), is a crucial immune regulator of inflammation known for its role in several pathological conditions, both infectious and due to sterile inflammation. IL-1R8 is highly conserved along the evolutionary scale and expressed by several cell types [4]. In particular, IL-1R8 is highly expressed in epithelial cells and lymphoid organs; differential expression patterns have been demonstrated in different animal species [5,6,7,8]. Multiple mechanisms of action have been uncovered for this receptor, which acts as a negative regulator of the signal transduction. It binds the powerful anti-inflammatory and immunosuppressive cytokine IL-37, which is induced by multiple pro-inflammatory stimuli and has numerous activities encompassing inhibition of pro-inflammatory cytokines and switch of immune cells from pro- to anti-inflammatory phenotype [9,10]. IL-37 shares similar structure with IL-18; therefore, the extracellular IL-37 can bind to IL-18 receptor alpha chain (IL-18Rα), creating a complex. Then, IL-1R8 ties up the molecular complex IL-37-IL-18Rα on the cell outward creating a tripartite complex, IL-37-IL-18Rα-IL-1R8, that transduces anti-inflammatory signals by the inhibition of NF-κB and MAPK and the activation of Mer-PTEN-DOK pathways [11]. Moreover, along with IL-1R5/IL-18Rα, IL-1R8 has been demonstrated to function as a co-receptor for the anti-inflammatory cytokine IL-37 [12]. Thus, it may be used as a marker for different types of immunopathologies. In addition to these mechanisms of action, IL-1R8 has been shown to reduce NF-κB and JNK activation and therefore block signaling transduction pathways downriver ILRs and TLRs via inhibition of MyD88 dimerization or retention of the Myddosome complex [13,14]. Moreover, the extracellular domain of IL-1R8 can inhibit the interaction between IL-1R1 and IL-1R3 [15] and TLR3 signaling via blockage of TRAM homodimerization and TRIF-TRAM interactions [14]. Finally, IL-1R8 is also involved in the regulation of the mTOR pathway in lymphoid and non-lymphoid cells [16].

IL-1R8 is highly expressed in the gut epithelium, where it tunes TLR reactivity against commensal bacteria; IL-1R8-deficient mice infected with *Citrobacter rodentium* displayed an exaggerated IL-1R1 signaling-dependent gut inflammation, causing a severe loss of commensal bacteria and facilitated secondary infection by *Salmonella typhimurium* [17]. Accordingly, the administration of probiotic bacteria for treatment of gut infections and diseases beneficially regulated host immune responses by modulating TLR negative regulators, including IL-1R8 [18,19].

Studies have shown that IL-1R8 is necessary for the anti-inflammatory potential of IL-37 in different pathologic conditions, including dextran sodium sulfate colitis [20]. Human and murine necrotizing enterocolitis has been proven to be characterized by dysregulated intestinal immune homeostasis and expression of TLR repertoires, particularly IL-1R8 and Il-37 are decreased [21].

Given the anti-inflammatory role of IL-1R8, its modulation in gastrointestinal diseases could represent a novel marker of diagnosis and prognosis.

After discovering the link between the spiral-shaped bacterium *Helicobacter pylori* and severe gastrointestinal diseases in humans [22], investigations also revealed high prevalence of bacteria of such genus in dogs, including gastric and enterohepatic *Helicobacter* spp. [23,24,25]. In humans, enterohepatic *Helicobacter* spp. have been identified in case of intestinal disorders and are considered to have a potential pathogenic role in the development of inflammatory bowel disease (IBD) [26,27,28,29]. However, in dogs although enterohepatic *Helicobacter* spp. infection has been reported, the potential relationship between *Helicobacter* spp. and CE remains unclear and poorly investigated [30]. Diagnosis of *Helicobacter* spp. in dogs is based on limited useful noninvasive or invasive tests [23]. Biomarkers of CE for diagnostic, therapeutic and prognostic purposes have not been investigated so far in dogs.

Diagnosis, control and treatment of canine gastrointestinal diseases, including the ones possibly due to *Helicobacter* spp. infection, are relevant in a One-Health as well as a veterinary approach. Dogs harboring *Helicobacter* not only need an early diagnosis and treatment to prevent disease progression, but represent also a potential zoonotic risk, being reservoirs of a human-affecting pathogen and posing public health implications [31,32]. Therefore, the search for novel markers potentially useful for diagnostic and therapeutic purposes is needed. We investigated the expression of the receptor IL-1R8 in colon biopsies of dogs with gastrointestinal signs and colonized by *Helicobacter* spp. in order to define its potential use in diagnostic, prognostic and potentially therapeutic protocols.

## 2. Materials and Methods

### 2.1. Samples

This study was performed on the same samples used for a previously published study [30]. Briefly, biopsies from the large intestines of 26 dogs of different breeds (13 males and 13 females) presenting intestinal or gastrointestinal signs were enrolled. In addition, 4 healthy controls (2 males and 2 females), without macroscopic lesions at the endoscopic exam, were included. Groups were defined according to gender, clinical condition (4 controls, 5 with acute and 21 with chronic form), *Helicobacter* spp. colonization (uncolonized, low colonized when biopsies were positive for *Helicobacter* spp. by PCR but low or scant amount of *Helicobacter* antigen in the intestinal crypts was detected by IHC and highly colonized when biopsies were positive for *Helicobacter* spp. by PCR and moderate to large amount of *Helicobacter* antigen in the intestinal crypts was detected by IHC) and age (0–1 years, 1–3 years, 3–8 years, > 8 years).

None of the animals had been treated with antibiotics 3 weeks prior examination. Endoscopy and biopsy sampling were performed as part of clinical investigation with the owner’s agreement. The study was approved by the Ethical Committee of the University of Milan (Protocol numbers 35/11 and 14/2022). Samples were taken from each animal for histological examination, PCR detection of *Helicobacter* spp. DNA, in situ analysis of *Helicobacter* spp. colonization and RNA extraction for IL-1R8 gene expression. Samples for histological examination were fixed in 10% neutral buffered formalin and embedded in paraffin wax; samples for PCR detection of *Helicobacter* spp. DNA were immediately frozen and stored at −20 °C; samples for RNA extraction were immediately placed in sterile tubes containing 1 mL of RNA Later (Qiagen, Hilden, Germany) and stored at 4 °C for 24 h.

Materials and methods related to PCR for genus determination and quantification of *Helicobacter* spp., histological examination for detection of lesions (hyperplasia, fibrosis, neutrophil infiltration and lymphocyte/plasma cell infiltration) and immunohistochemistry for in situ analysis of *Helicobacter* spp. Colonization, along with record of endoscopic lesions, are described in detail by Castiglioni et al. (2012) [30].

### 2.2. RNA Extraction

Total RNA was isolated from the samples by the guanidine isothiocyanate method with minor modifications. Briefly, the samples were homogenized in 1 mL of guanidine isothiocyanate (4 M) using a rotor-stator system (Ultra Turrax T25 Ika-Werke, Staufen, Germany). The lysate was ultracentrifuged overnight at 113,000× *g* at 18 °C on a 5.7 M cesium chloride layer (Optima TL ultracentrifuge, Beckman Instruments, Inc., Palo Alto, CA, USA). The RNA pellet was dissolved in sterile water and precipitated with absolute ethanol and sodium acetate 3 M pH 5.4 in dry ice for 2 h. After centrifugation in a microcentrifuge (Eppendorf, Hamburg, Germany) at 13,000× *g* for 30 min, the RNA pellet was dissolved in sterile water and stored at −20 °C. The concentration of RNA was determined using a spectrophotometer (BioPhotometer, Eppendorf, Hamburg, Germany) at 260 nm wavelength.

### 2.3. Reverse Transcription and Real-Time PCR

Total RNA (1 μg) of each sample was reverse transcribed to cDNA using the High Capacity cDNA Archive kit with random hexamers (Applied Biosystem, Foster City, CA, USA), and the resulting cDNA was stored at −20 °C before real-time PCR assays.

The cDNA obtained from each sample was used as a template for real-time PCR in an optimized 25-μL reaction volume in MicroAmp optical 96-well plates. Each plate contained duplicates of each cDNA sample, 2X Power Syber Green PCR Master Mix (Applied Biosystem, Foster City, CA, USA) and primers at 300 nM each. The species-specific primer pairs were designed by Primer Express using as target the dog sequence homologous to human and mouse IL-1R8 available in the NCBI nucleotide sequences database with accession number XM_038423988.1 (Gene Bank). Dog IL-1R8 primer sequences are: forward 5′-CTACACATCCTCCTCAGACAC-3′ and reverse 5′-GTGTCCGAGGGCCTATCTTTG-3′. To correct for variations of extracted mRNA amounts and cDNA synthesis efficacy in quantitative real-time PCR assays, primers for the detection of the dog housekeeping gene beta-actin were designed as well by Primer Express on the sequence accession number AF021873.2 (Gene Bank), forward 5′-TCCCTGGAGAAGAGCTACGA-3′ and reverse 5′-CTTCTGCATCCTGTCAGCAA-3′. All primers were custom synthesized by Invitrogen (Carlsbad, CA, USA). A duplicate no-template control (NTC) was also included in each plate. Real-time quantitative PCR was carried out in the 7000 Sequence Detection System (Applied Biosystem, Foster City, CA, USA) at the following thermal cycle conditions, 10 min at 95 °C followed by 40 cycles of 15 s at 95 °C and 1 min at 60 °C. Quantification was determined after application of an algorithm to the data analyzed by the software of the 7000 Detection System (Applied Biosystem, Foster City, CA, USA). The expression of IL-1R8 of each biopsy was normalized using the calculated beta-actin cDNA expression (mean) of the same sample and run.

### 2.4. Software

We used the free software BLAST available on the PubMed website for gene sequence search and alignment. Primer Express software (Applied Biosystem, Foster City, CA, USA) was used for the specific primer design. Statistical analysis was made using GraphPad Prism 6 (La Jolla, CA, USA) software.

### 2.5. Statistical Analysis

Statistical analyses were performed considering statistically significant values at *p* < 0.05 and tendencies at *p* < 0.1. Shapiro-Wilk tests and diagnostic graphics were used to verify the distribution of data and the equality of variances. The expression of IL-1R8 gene in two experimental groups (females vs. males) was compared by Mann-Whitney test. Finally, two-tails rho tests of Spearman were used to reveal correlations among target gene expression and all the other parameters.

The Kruskal-Wallis test was used to compare the expression level of IL-1R8 in the 3 different groups based on type of pathological condition (control, acute and chronic form and uncolonized, low colonized and highly colonized) and 4 age groups (0–1 years, 1–3 years, 3–8 years, > 8 years).

## 3. Results

### 3.1. IL-1R8 Expression Is Not Influenced by Sex and Age

IL-1R8 gene expression was compared among groups, and no difference was observed between males and females, or between young and old dogs (Table 1 and Figure 1).

### 3.2. IL-1R8 Is Downregulated during Inflammation

Dogs enrolled in the study were divided in groups based on the clinical condition: control (4 dogs), acute inflammation (5 dogs; clinical signs present since less than three weeks), chronic inflammation (21 dogs; clinical signs present since more than three weeks) [2]. We observed a significant downregulation of the expression of IL-1R8 mRNA in acute and chronic inflamed samples compared to controls (Figure 2).

### 3.3. The Level of Helicobacter Colonization Modulates the Expression of IL-1R8 in the Gut

Samples were classified into three classes (i.e., heavily colonized, poorly colonized, and uncolonized biopsies) according to their *Helicobacter* spp. colonization status based on PCR and immunohistochemical results as previously described [30]. The expression of IL-1R8 resulted gradually reduced from uncolonized to poorly colonized and to highly colonized samples (Figure 3a), although the difference was not statistically significant. This seems to be confirmed by the downregulation of IL-1R8 mRNA in samples with bacteria infiltration into the crypts (Figure 3b; not statistically significant) and significant negative correlation between IL-1R8 and colonization or PCR positivity for *Helicobacter* (Table 2).

### 3.4. The Expression Level of IL-1R8 mRNA Correlates with Hyperplasia

Finally, we analyzed the possible correlation among IL-1R8 gene expression and the previously reported results of all the other parameters investigated (breed, sex, age, endoscopic lesions, hyperplasia score, fibrosis, neutrophil infiltration, lymphocyte/plasma cell infiltration, *Helicobacter* genus, level of colonization) (Table 2). Interestingly, IL-1R8 expression negatively correlated with the level of colonization and the presence of macroscopic lesions observed by endoscopy (*p* = 0.0399 and *p* = 0.0192 respectively). A correlation was also observed with the breed of dogs (*p* = 0.0079), but the most significant correlation was a negative one with hyperplasia score (*p* < 0.0001). No correlation was identified between IL-1R8 gene expression and the other histological parameters considered (sex, age, fibrosis, neutrophil infiltration, lymphocyte/plasma cell infiltration, *Helicobacter* genus).

## 4. Discussion

The epithelium of the gastrointestinal tract is in strict contact with microorganisms that could continuously activate an inflammatory response that could become detrimental for the organism. To avoid this, fine mechanisms negatively regulate pro-inflammatory pathways in the gut. When the epithelium is damaged or the pathogen bacteria colonize and invade the mucosa, the inflammatory response is activated to fight the aggressor. Some gastrointestinal diseases are associated with prolonged and uncontrolled activation of inflammatory response, finally leading to tissue damage. Il-1R8 is known to act as a negative regulator of NF-κB and JNK activation following stimulation of IL-1R family members or TLRs. IL-1R8 is highly expressed in intestinal epithelium in order to keep under control the commensal microflora stimulation and ensure immune tolerance to the microbiota but response to pathogens. Therefore, we focused the study on the expression of IL-1R8 in large intestine biopsies of dogs infected with *Helicobacter* spp. and uninfected controls.

We showed that IL-1R8 mRNA expression is downregulated during both acute and chronic inflammation compared to control tissues. Our results are in agreement with previous data published by Anselmo et al. 2016, Li et al. 2015 and Veliz-Rodriguez et al. 2012, which showed that different inflammatory conditions are associated to reduction of IL-1R8 gene expression [13,31,32] due to the negative regulation of SP1 binding to IL-1R8 promoter [33,34].

In our data both acute and chronic enteropathies lead to a similar downregulation of IL-1R8, but the level of enterohepatic *Helicobacter* infection cause different levels of downregulations. Indeed, we observed a progressive reduction of IL-1R8 expression in poorly colonized and heavily colonized samples as compared to uncolonized ones. In particular, the presence of *Helicobacter* spp. within the crypts surface causes an evident down regulation of IL-1R8. Indeed, the presence of crypt colonization is considered a real *Helicobacter* infection, whereas the only presence of *Helicobacter* on the epithelium surface could be due to the transit of bacteria coming from the stomach [30].

Interestingly, the expression of IL-1R8 is not affected by sex and age. On the contrary, another study demonstrated a downregulation of IL-1R8 in old rats [35]. The difference of results could be due the fact that, in our study, we could not enroll very old dogs (maximum 12 years), whereas in the study by Xu et al., they compared very young rats (3 months) with very old rats (24 months) [35].

We also described a negative correlation between the level of expression of IL-1R8 and the presence of macroscopic lesions identified by endoscopy and the hyperplasia score. Indeed, macroscopic lesions are associated with higher levels of inflammation, and inflammation can promote hyperplasia.

An association was uncovered between IL-1R8 expression level and dog breed, which could be related to specific polymorphisms of the gene in different populations of dogs [36,37].

In the intestine, IL-1R8 generally exerts protective functions; however, IL-1R8 activity is dependent on the cytokine milieu [38]. In other tissues, IL-1R8 deficiency is associated with protection against pathogen infection [39,40].

In this setting, different probiotic bacteria utilized as a remedy for pathological conditions of the intestine function by tuning negative regulators of TLR and ILR and boost the host immune response [18,19].

Taken together, our observations strongly suggest that a deficiency in IL-1R8 contributes to CE pathogenesis. As pathology advances, receptor disappears, thus likely accelerating the vicious circle of excessive inflammation and intestinal injury. This is in agreement with what was observed in the model of human necrotizing enterocolitis reported by Cho et al. [21].

## 5. Conclusions

We demonstrated an important role of IL-1R8 in canine chronic enteropathy. Our preliminary study showed that IL-1R8 could represent a marker of the clinical severity of the gut disease and is modulated by *Helicobacter* spp. colon infection. Moreover, IL-1R8 could be a target of new therapeutic strategies that aim to increase its expression level. In view of all of this, a number of probiotic bacteria may be assayed in dogs infected by *Helicobacter*.

IL-1R8 thus represents an intriguing pathway marker that merits exploration in terms of modulating responses against autoantigens and alloantigens in the intestine. Our study may have significance for the improvement of diagnostic, therapeutic and prognostic approaches.

## Figures and Tables

**Figure 1 vetsci-09-00295-f001:**
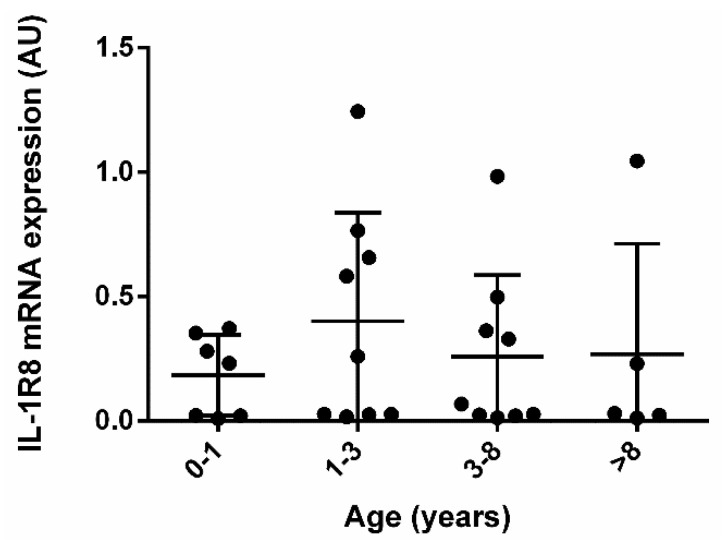
**IL-1R8 mRNA expression in samples from dogs of different age.** IL-1R8 mRNA expression was analyzed by real-time PCR in colon biopsies of dogs with gastrointestinal signs. The gene expression level of the target gene was normalized to beta-actin gene expression and the results are presented as arbitrary Units (2^−^^ΔCt^ × 10,000). Dogs enrolled in the study were arbitrarily divided into 4 age classes (Kruskal-Wallis test *p* = 0.6878).

**Figure 2 vetsci-09-00295-f002:**
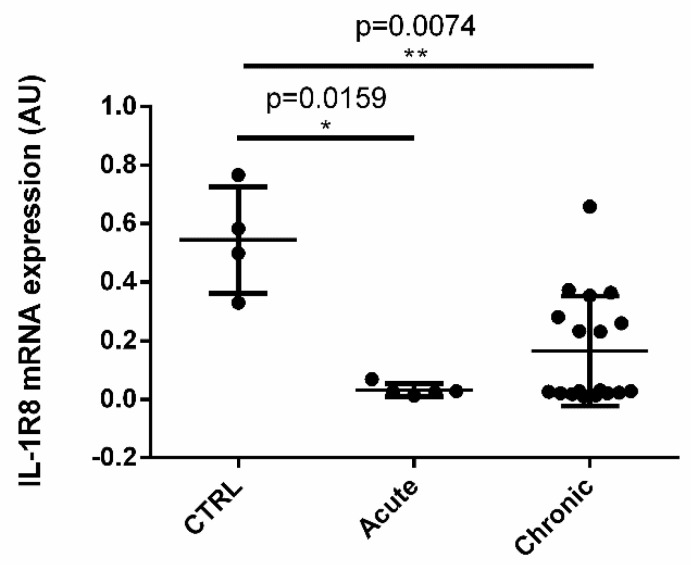
**IL-1R8 mRNA expression by real-time PCR in control, acute and chronic inflamed samples.** The gene expression level of the target gene was normalized to beta-actin gene expression and the results are presented as Arbitrary Unit (2^−^^ΔCt^). Dogs enrolled in the study were divided into 3 groups based on the clinical condition (Kruskal-Wallis test *p* = 0.0145; Mann-Whitney test) (* *p* ˂ 0.05; ** *p* ˂ 0.01).

**Figure 3 vetsci-09-00295-f003:**
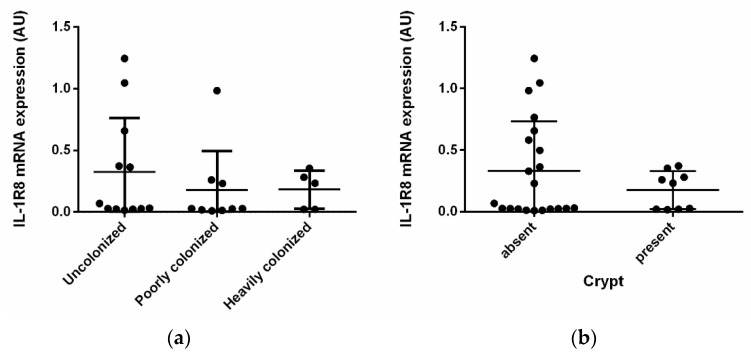
**IL-1R8 mRNA expression in *Helicobacter* infected samples.** IL-1R8 mRNA expression was analyzed by real-time PCR in colon biopsies of dogs with different levels of *Helicobacter* colonization. The gene expression level of the target gene was normalized to beta-actin gene expression and the results are presented as Arbitrary Unit (2^−^^ΔCt^). (**a**) Dogs enrolled in the study were divided into 3 groups based on the level of *Helicobacter* colonization (Kruskal-Wallis test *p* = 0.4080); (**b**) dogs enrolled in the study were divided into 2 groups based on the presence or absence of *Helicobacter* into the crypts (Student’s *t* test *p* = 0.267).

**Table 1 vetsci-09-00295-t001:** IL-1R8 gene expression in females and males.

	Females	Males	*p* Value
**Average IL-1R8 expression (AU)**	0.2144	0.2773	0.41

Mann-Whitney test.

**Table 2 vetsci-09-00295-t002:** **Correlation among IL-1R8 gene expression and other parameters**.

	Parameter	R Spearman	*p* Value
**IL-1R8 expression**	Dog breed	0.3398	0.0079
	Endoscopy	−0.3178	0.0192
	Hyperplasia	−0.4875	<0.0001
	Helicobacter int	−0.2927	0.0232
	Colonization	−0.2661	0.0399

## Data Availability

Not applicable.

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
