# Peer review of "IL-1R8 as Pathoimmunological Marker for Severity of Canine Chronic Enteropathy"

_vetsci, 2022, doi:10.3390/vetsci9060295_

Round 1

Reviewer 1 Report

The paper describes chronic enteropathy (CE) that affects dogs and is driven by poorly characterized inflammatory pathways. The authors showed that the expression of IL-1R8 is downregulated in both acutely and chronically CE affected dogs compared to healthy controls. A negative correlation was detected between the level of expression of IL-1R8 and the severity of macroscopic lesions identified by endoscopy and the crypt hyperplasia score. The data suggest the advance knowledge on gastrointestinal pathoimmunology in dogs and highlight the potential utilization of IL-1R8 as diagnostic, prognostic and therapeutic biomarker for canine chronic enteropathy.

Author Response

We agree.

Reviewer 2 Report

The article is original and very interesting. I suggest some small corrections.

1. In Section 3.4 of Results I suggest to introduce some Immunohistochemistry lesions to show lymphoid infiltrations, degree of inflammation/ hyperplasia

2.lines 307-308 delete Instructions for author Please turn to the CRediT taxonomy for the term explanation. Authorship must
308 be limited to those who have contributed substantially to the work reported.

Author Response

Author's Reply to the Review Report (Reviewer 2)

Comments and Suggestions for Authors

The article is original and very interesting. I suggest some small corrections.

  1. In Section 3.4 of Results I suggest to introduce some Immunohistochemistry lesions to show lymphoid infiltrations, degree of inflammation/hyperplasia

We thank the reviewer for the suggestion, but since the histological results and representative images are already reported in the previous paper from Castiglioni et al., 2012, which we cited as reference [30] (previous reference [33]), we think it is not appropriate to report here the already published results. To clarify this point, we rephrased the statement in section 3.4. at previous lines 224-227, now lines 240-243 as “Finally, we analyzed the possible correlation among IL-1R8 gene expression and the previously reported results of all the other parameters investigated (breed, sex, age, endoscopic lesions, hyperplasia score, fibrosis, neutrophil infiltration, lymphocyte/plasmacell infiltration, Helicobacter genus, level of colonization)”.

2.lines 307-308 delete Instructions for author Please turn to the CRediT taxonomy for the term explanation. Authorship must be limited to those who have contributed substantially to the work reported.

At previous lines 307-308 (now line 321) we removed the part of sentence that had been left from the Instructions for the authors.

Reviewer 3 Report

Dear authors,

Keywords: IL-1R8; chronic enteropathy;  Helicobacter spp.; dog, biomarker.

Introduction

Lines 32 to 34 - Canine chronic enteropathy (CE) is an intestinal idiopathic inflammatory disorder which has gained considerable attention in recent years due to severity of symptoms, failure of antibiotic- and drug-based treatments and irreversible damage outcome.

Antibiotics are not drugs?

Do you mean diet, antibiotics and immunosuppressants?

Lines 34 to 36 - Endoscopic and histologic examinations are still the gold standard methods for diagnosis and determination of the extent of the disease. 

I'm afraid that the notion of "Canine chronic enteropathy" conveyed in your work is not clear. It would be useful to refer to the various types of CCE as responsive to diet, antibiotics or immunosuppressants and a clear relationship with the term IBD.

Endoscopic and histopathological analysis of biopsy samples are related to the concept of IBD.

Line 37 - objective assessment of CE diagnosis, severity and prognosis

Line 39 - Please pre-set all the text abbreviations as ...Single Ig IL-1-related receptor (SIGIRR), also called Toll/Interleukin-1 receptor 8 (TIR8) or Interleukin-1 receptor 8 (IL-1R8),

Line 88 - Biomarkers for diagnostic, therapeutic and prognostic purposes

In my opinion, the team of authors support a clearly unproven line of reasoning (relationship Helicobacter pylori and inflammatory colon disease in the dog), using references 24 to 30 from human studies.

Although Helicobacter pylori is a recognized pathogen of the human species, even a WHO-recognized carcinogenic, it is not present in the dog, nor is it clear that other elements of this genus (Helicobacter like spp ) present in the dog (stomach or colon) are pathogens of this species.

It is also not commonly accepted that the dog can be a reservoir of these species and that its relationship with humans can be the key to human infection.

In the cited bibliographic pieces (31 and 32) there is no support for the idea that Helicobacter is the etiological agent of chronic enteropathy in dogs.

Material and Methods

Line 105 - at the endoscopic exam.

Results

Line 190 - Figure 1 - The division of the sample into age groups must be previously mentioned in the Material and Methods section.

As an infection is the invasion of an organism's body tissues by pathogens, their multiplication, and the reaction of host tissues to the infectious agents and the toxins they produce, the use of the term infection in the legend of this figure does not seem clear to me.

Line 195 to 197 - Dogs enrolled in the study were divided in groups based on the clinical condition: control (4 dogs), acute inflammation (5 dogs; clinical signs present since less than three weeks), chronic inflammation (21 dogs; clinical signs present since more than three weeks) [2].

The information underlined in yellow should be provided before in the MM section.

Why do you put here reference 2?

Lines  206 to 207 - Samples were classified into three classes (i.e. heavily colonized, poorly colonized, and uncolonized biopsies) according to their Helicobacter spp. colonization status based on PCR and immunohistochemical results as previously described [33].

Although referring to reference 33, the description of these classes in section MM would facilitate the understanding of the work.

Lines 225 to 227 - investigated parameters should be previously described in MM section [breed, sex, age, macroscopic endoscopic lesions, Histopathological assessment of biopsy samples (hyperplasia score, fibrosis, neutrophil infiltration, lymphocyte/plasma cell infiltration), Helicobacter genus presence and level of colonization].

Line 229  - A correlation was also observed with the breed of dogs (Please describe the refered breeds)

Lines 231 to 233 - No correlation was identified between IL-1R8 gene expression and the other histological parameters (sex, age, fibrosis, neutrophil infiltration, lymphocyte/plasmacell infiltration, Helicobacter genus). 

Sex, age and Helicobacter genus are histological parameters?

Discussion

Line 253 - Anselmo et al. 2016, Veliz-Rodriguez et al. 2012 and Li et al. 2015,

They are not described alphabetically or chronologically!

It is essential when we talk about TLR to refer to the term immunotolerance, especially in the context of gastrointestinal disease.

Conclusions

Lines 297 to 299 - Our study may have implications for the development of diagnostic, therapeutic and  prognostic approaches.

Author Response

Author's Reply to the Review Report (Reviewer 3)

Comments and Suggestions for Authors

Keywords: IL-1R8; chronic enteropathy; Helicobacter spp.; dog, biomarker.

According to the advice, we removed “IBD; inflammatory bowel disease” from the keywords and we added “biomarker”, but we would leave “TIR8” and “SIGIRR” because of a very recent re-nomination of the receptor and according to what the other publications on the specific topic do.

Introduction

Lines 32 to 34 - Canine chronic enteropathy (CE) is an intestinal idiopathic inflammatory disorder which has gained considerable attention in recent years due to severity of symptoms, failure of antibiotic- and drug-based treatments and irreversible damage outcome.

Antibiotics are not drugs? Do you mean diet, antibiotics and immunosuppressants?

At previous line 34 (now line 33) the statement has been rephrased as “failure of diet-, antibiotic- and other drug-based treatments”

Lines 34 to 36 - Endoscopic and histologic examinations are still the gold standard methods for diagnosis and determination of the extent of the disease. I'm afraid that the notion of "Canine chronic enteropathy" conveyed in your work is not clear. It would be useful to refer to the various types of CCE as responsive to diet, antibiotics or immunosuppressants and a clear relationship with the term IBD. Endoscopic and histopathological analysis of biopsy samples are related to the concept of IBD.

According to the comment, for clarity we replaced at previous lines 34-36 (now lines 34-37) “Endoscopic and histologic examinations are still the gold standard methods for diagnosis and determination of the extent of the disease. However” with “CE includes food-responsive, antibiotic-responsive, immunosuppressant-responsive and non-responsive cases [2]. Diagnosis of CE includes endoscopic and histologic examinations of biopsies after other all other possible causes of disease have been excluded. Therefore”.

Line 37 - objective assessment of CE diagnosis, severity and prognosis

At previous line 37 (now line 38) we agreed to replace the words as suggested.

Line 39 - Please pre-set all the text abbreviations as ...Single Ig IL-1-related receptor (SIGIRR), also called Toll/Interleukin-1 receptor 8 (TIR8) or Interleukin-1 receptor 8 (IL-1R8),

We specified all the abbreviations. At previous line 39 (now line 39) we added “(Interleukin-1)” and “(Interleukin-1 Receptor 8)”; at now lines 40-41 we added “(Toll/Interleukin-1 Receptor 8)” and “(Single-Immunoglobulin-Interleukin-1 Related Receptor)”.

Line 88 - Biomarkers for diagnostic, therapeutic and prognostic purposes

At previous line 88, now line 89 we corrected according to the suggestion.

In my opinion, the team of authors support a clearly unproven line of reasoning (relationship Helicobacter pylori and inflammatory colon disease in the dog), using references 24 to 30 from human studies. Although Helicobacter pylori is a recognized pathogen of the human species, even a WHO-recognized carcinogenic, it is not present in the dog, nor is it clear that other elements of this genus (Helicobacter like spp ) present in the dog (stomach or colon) are pathogens of this species.

We extensively revised this part of the introduction to better elucidate the background information about Helicobacter, as indicated by the reviewer.

The statements at previous lines 78-89 have been replaced with the following statements: “After discovering the link between the spiral-shaped bacterium Helicobacter pylori and severe gastrointestinal diseases in humans [22], investigations revealed high prevalence of bacteria of such genus also in dogs, including gastric and enterohepatic Helicobacter spp. [23-25]. In humans, enterohepatic Helicobacter spp. have been identified in case of intestinal disorders and are considered to have a potential pathogenic role in the development of inflammatory bowel disease (IBD) [26-29]. However, in dogs although enterohepatic Helicobacter spp. infection has been reported, the potential relationship between Helicobacter spp. and CE remains unclear and poorly investigated [30]. Diagnosis of Helicobacter spp. in dogs is based on limited useful noninvasive or invasive tests [23]. Biomarkers of CE for diagnostic, therapeutic and prognostic purposes have not been investigated so far in dogs.”

Accordingly, also the list of references was updated.

The following references were added:

  • [24] Recordati C., Gualdi V., Craven M., Sala L., Luini M., Lanzoni A., Rishniw M., Simpson K.W., Scanziani E. Spatial Distribution of Helicobacter in the Gastrointestinal Tract of Dogs. Helicobacter. 2009,14:180-191.
  • [25] Jalava K, On SL, Vandamme PA, Happonen I, Sukura A, Hänninen ML. Isolation and identification of Helicobacter from canine and feline gastric mucosa. Appl Environ Microbiol 1998;64:3998–4006.
  • [26] Castaño-Rodríguez N, Kaakoush NO, Lee WS, Mitchell HM. Dual role of Helicobacter and Campylobacter species in IBD: a systematic review and meta-analysis. Gut. 2017 Feb;66(2):235-249. doi: 10.1136/gutjnl-2015-310545. Epub 2015 Oct 27. PMID: 26508508.
  • [28] Tankovic, J., Smati, M., Lamarque, D., Delchier, J.C., 2011. First detection of Helicobacter canis in chronic duodenal ulcerations from a patient with Crohn’s disease. Inflamm. Bowel Dis. 17, 1830–1831.
  • [29] Thomson, J.M., Hansen, R., Berry, S.H., Hope, M.E., Murray, G.I., Mukhopadhya, I., McLean, M.H., Shen, Z., Fox, J.G., El-Omar, E., Hold, G.L., 2011. Enterohepatic Helicobacter in ulcerative colitis: potential pathogenic entities? PLoS One 6, 1–9.
  •  

The following references were deleted:

  • [24] Osaki, T.; Zaman, C.; Yonezawa, H.; Lin, Y.; Okuda, M.; Nozaki, E.; Hojo, F.; Kurata, S.; Hanawa, T.; Kikuchi, S.; Kamiya, S. Influence of intestinal indigenous microbiota on intrafamilial infection by Helicobacter pylori in Japan. Immunol. 2018; 9, 287.
  • [25] Dash, N.R.; Khoder, G.; Nada, A.M.; Al Bataineh, M.T. Exploring the impact of Helicobacter pylori on gut microbiome composition. PLoS One. 2019, 14, e0218274.
  • [26] Heimesaat, M.M.; Fischer, A.; Plickert, R.; Wiedemann, T.; Loddenkemper, C.; Göbel, U.B.; Bereswill, S.; Rieder, G. Helicobacter pylori induced gastric immunopathology is associated with distinct microbiota changes in the large intestines of long-term infected Mongolian gerbils. PLoS One. 2014, 9, e100362.
  • [27] Bartels, L.E.; Jepsen, P.; Christensen, L.A.; Gerdes, L.U.; Vilstrup, H.; Dahlerup, J.F. Diagnosis of Helicobacter pylori infection is associated with lower prevalence and subsequent incidence of Crohn's disease. Crohns Colitis. 2016, 10, 443-448.
  • [28] Shirzad-Aski, H.; Besharat, S.; Kienesberger, S.; Sohrabi, A.; Roshandel, G.; Amiriani, T.; Norouzi, A.; Keshtkar, A. Association between Helicobacter pylori colonization and inflammatory bowel disease: a systematic review and meta-analysis. Clin. Gastroenterol. 2021, 55, 380-392.
  • [29] Guo, Y.; Xu, C.; Gong, R.; Hu, T.; Zhang, X.; Xie, X.; Chi, J.; Li, H.; Xia, X.; Liu, X. Exosomal CagA from Helicobacter pylori aggravates intestinal epithelium barrier dysfunction in chronic colitis by facilitating claudin-2 expression. Gut Pathog. 2022, 14, 13.

We re-numbered references starting from [24] in both the text and the Reference sections.

It is also not commonly accepted that the dog can be a reservoir of these species and that its relationship with humans can be the key to human infection. In the cited bibliographic pieces (31 and 32) there is no support for the idea that Helicobacter is the etiological agent of chronic enteropathy in dogs.

Please see the answer to the comment above.

Material and Methods

Line 105 - at the endoscopic exam.

At previous line 105 (now line 107) as suggested we replaced “endoscopic analysis” with “endoscopic exam”.

Results

Line 190 - Figure 1 - The division of the sample into age groups must be previously mentioned in the Material and Methods section.

In agreement with the comment, we specified all the groups in Materials and Methods section “Samples” (previous line 106, now lines 107-113).

As an infection is the invasion of an organism's body tissues by pathogens, their multiplication, and the reaction of host tissues to the infectious agents and the toxins they produce, the use of the term infection in the legend of this figure does not seem clear to me.

We thank the referee for the comment and we corrected the legend of Figure 1 at previous line 190 (now line 199) replacing “affected by Helicobacter infection” with “with gastrointestinal signs.”.

In addition, we replaced the term “infection” with “colonization” at previous line 97 (now line 98) of the Introduction section and at previous line 205 (low line 214) in the title of chapter 3.3.

Line 195 to 197 - Dogs enrolled in the study were divided in groups based on the clinical condition: control (4 dogs), acute inflammation (5 dogs; clinical signs present since less than three weeks), chronic inflammation (21 dogs; clinical signs present since more than three weeks) [2].

The information underlined in yellow should be provided before in the MM section. Why do you put here reference 2?

According also to the second last above comment of the same referee, we specified all the groups in Materials and Methods section “Samples” (previous line 106, now lines 107-113). We removed reference [2].

Lines 206 to 207 - Samples were classified into three classes (i.e. heavily colonized, poorly colonized, and uncolonized biopsies) according to their Helicobacter spp. colonization status based on PCR and immunohistochemical results as previously described [33].

Although referring to reference 33, the description of these classes in section MM would facilitate the understanding of the work.

As suggested, we added in Materials and Methods section “Samples” the description of these classes (previous lines 206-207 (now lines 107-113).

Lines 225 to 227 - investigated parameters should be previously described in MM section [breed, sex, age, macroscopic endoscopic lesions, Histopathological assessment of biopsy samples (hyperplasia score, fibrosis, neutrophil infiltration, lymphocyte/plasma cell infiltration), Helicobacter genus presence and level of colonization].

As suggested, we indicated in Materials and Methods section “Samples” all the investigated parameters (previous lines 217-120, now lines 126-129) and we rephrased the statement in section 3.4. at previous lines 224-227, now lines 240-241 (please see the answer to the first comment of Reviewer 2).

Line 229 - A correlation was also observed with the breed of dogs (Please describe the refered breeds)

We indicated at previous line 103 (now line 104) of Materials and Methods section that different breeds of dogs were included.

Lines 231 to 233 - No correlation was identified between IL-1R8 gene expression and the other histological parameters (sex, age, fibrosis, neutrophil infiltration, lymphocyte/plasmacell infiltration, Helicobacter genus). Sex, age and Helicobacter genus are histological parameters?

We corrected the mistake at previous lines 231-233 (now lines 248-249) by replacing “histological” with “considered”.

Discussion

Line 253 - Anselmo et al. 2016, Veliz-Rodriguez et al. 2012 and Li et al. 2015,

They are not described alphabetically or chronologically!

At previous line 253 (now line 267) we reported them alphabetically.

It is essential when we talk about TLR to refer to the term immunotolerance, especially in the context of gastrointestinal disease.

At previous line 248 (now lines 262-263) we referred to the term “immune tolerance”,

Conclusions

Lines 297 to 299 - Our study may have implications for the development of diagnostic, therapeutic and prognostic approaches.

At previous lines 297-299, now lines 312-313 we corrected as advised.

Round 2

Reviewer 3 Report

I am grateful for all the work developed by the authors in order to improve the final version of the work. Congratulations!